# Comparison of two cable configurations in 3D printed steerable instruments for minimally invasive surgery

**Costanza Culmone**[1]*, **Remi van Starkenburg**[2], **Gerwin Smit**[1], **Paul Breedveld**[1]

**1** Faculty of Mechanical, Maritime, and Materials Engineering, Department BioMechanical Engineering, Bio-Inspired Technology Group (BITE), Delft University of Technology, Delft, The Netherlands, **2** Dienst Elektronische en Mechanische Ontwikkeling, Delft University of Technology, Delft, The Netherlands

* c.culmone@tudelft.nl

**Data Availability Statement:** All the data collected during the test are available from the 4TU. ResearchData repository database DOI: https://doi.org/10.4121/20497185.v1.

## Abstract

In laparoscopy, a small incision size improves the surgical outcome but increases at the same time the rigidity of the instrument, with consequent impairment of the surgeon's maneuverability. Such reduction introduces new challenges, such as the loss of wrist articulation or the impossibility of overcoming obstacles. A possible approach is using multi-steerable cable-driven instruments fully mechanical actuated, which allow great maneuverability while keeping the wound small. In this work, we compared the usability of the two most promising cable configurations in 3D printed multi-steerable instruments: a parallel configuration with all cables running straight from the steerable shaft to the handle; and a multi configuration with straight cables in combination with helical cables. Twelve participants were divided into two groups and asked to orient the instrument shaft and randomly hit six targets following the instructions in a laparoscopic simulator. Each participant carried out four trials (two trials for each instrument) with 12 runs per trial. The average task performance time showed a significant decrease over the first trial for both configurations. The decrease was 48% for the parallel and 41% for the multi configuration. Improvement of task performance times reached a plateau in the second trial with both instruments. The participants filled out a TLX questionnaire after each trial. The questionnaire showed a lower burden score for the parallel compared to multi configuration (23% VS 30%). Even though the task performance time for both configurations was comparable, a final questionnaire showed that 10 out of 12 participants preferred the parallel configuration due to a more intuitive hand movement and the possibility of individually orienting the distal end of the steerable shaft.

## 1. Introduction

Laparoscopic surgery is a minimally invasive procedure in which several small incisions allow access to the human body by means of long and straight surgical tools. The reduction of the incision size reduces the post-operative pain and the recovery time for the patient, minimizes the scar tissue, thus obtaining better cosmetic results, and improves the cost-effectiveness of

**Funding:** This work was supported by Netherlands Organization for Scientific Research (Nederlandse Organisatie voor Wetenschappelijk Onderzoek, NWO), domain Applied and Engineering Sciences (TTW), and which is partly funded by the Ministry of Economic Affairs. Grant number 12137, Bio-Inspired Maneuverable Dendritic Devices for Minimally Invasive Surgery, awarded to PB. URL: https://www.nwo.nl/. The funders had no role in the study design, data collection and analysis, or preparation of the manuscript. There was no additional external funding received for this study.

**Competing interests:** The authors have declared that no competing interests exist.

the procedure. Despite its great advantages, laparoscopy introduces new hurdles, e.g., due to the loss of wrist articulation and the introduction of a fulcrum effect [1,2]. Due to the pivoting point in the abdominal wall, the movement of the end-effector is inverted with respect to the handle. This so-called fulcrum effect results in a steeper learning curve. With the advent of new domains of minimally invasive surgery, such as single-port laparoscopy, transluminal, and intraluminal procedures, new challenges arise. For instance, accessing the target area becomes demanding when its optimal approach direction is not aligned with the rigid instrument shaft inserted through the incision [3].

Many robotic platforms have been proposed to overcome the limits in laparoscopy. One of the most famous platforms is the Da Vinci® robotic system offered by Intuitive Surgical Inc. (Intuitive Surgical Inc., Sunnyvale, Ca, USA) [4]. Robotic platforms give the surgeon additional degrees of freedom (DOF), three-dimensional visualization of the surgical site, and eliminate the fulcrum effect. However, they require a large footprint and high maintenance cost that makes the price-benefit ratio unfavorable for many procedures [5].

An alternative approach is the use of handheld mechanical solutions, in which the surgeon's dexterity is enhanced by a steering mechanism with an additional two DOF close to the end-effector. Many research prototypes and commercialized instruments have been designed, and different solutions have been proposed to control the steerability of the end-effector [6,7]. The two most used control strategies in handheld instruments are *wrist control*, in which the movement of the wrist is used to steer the end-effector, such as found in the Laparo-Angle [8] or the LaparoFlex [9], and *thumb control*, in which the thumb controls the steering by means of a joystick [10], a trackball [11], or a steering wheel [12,13]. Comparative studies have been carried out on these two different control strategies to identify the most beneficial handheld control for the surgeon [14–16]. However, despite the 2-DOF steerable end-effector, the shaft rigidity of these instruments still restricts the surgeon's workspace, limiting surgical use to procedures in which no obstacles need to be passed without being touched. To further improve maneuverability, mechanical solutions such as cable-driven mechanisms [17–19] or continuum concentric tubes [20] have been proposed to design a multi-steerable shaft enabling the surgeon to move along complex double-curved paths. Cable-driven solutions represent a valid alternative to robotic solutions due to their low maintenance cost, low noise, high sensitivity, and speed. Moreover, they directly react to the surgeon's movements providing direct feedback and they enable simplification of the design without compromising the instrument functionalities. In cable-driven solutions, the *cable control strategy* plays an important role [21]. Cables can vary from a minimum of three for steering in two planes [22] to four or more as in the so-called cable-ring configuration [23], Fig 1A. In our group, we have explored two different cable control strategies for controlling cable-driven multi-steerable instruments: a control strategy based on cables straightly guided from the steerable shaft to the control handle (*parallel configuration*) [24], and a control strategy based on the combination of straight and helically cables placed around the backbone of the shaft and the control handle (*multi configuration*) [17], Fig 1B and 1C.

Whereas control strategy comparisons have been performed for 2-DOF instruments with only one steerable segment, a comparison in the steering and control of multi-steerable instruments with two or more segments has not yet been carried out. As a result, there is a lack of information about which way of controlling multi-steerable instruments is more convenient to the surgeon. In this study, we developed 3D printed multi-steerable instruments using parallel and multi configurations. The study aims to highlight and compare important weak and strong point of the new designs, that can help to improve the design of future multi-steerable laparoscopic instruments. For this purpose, using these instruments, we carried out an experiment with 12 participants to compare the two control strategies and identify which one has a

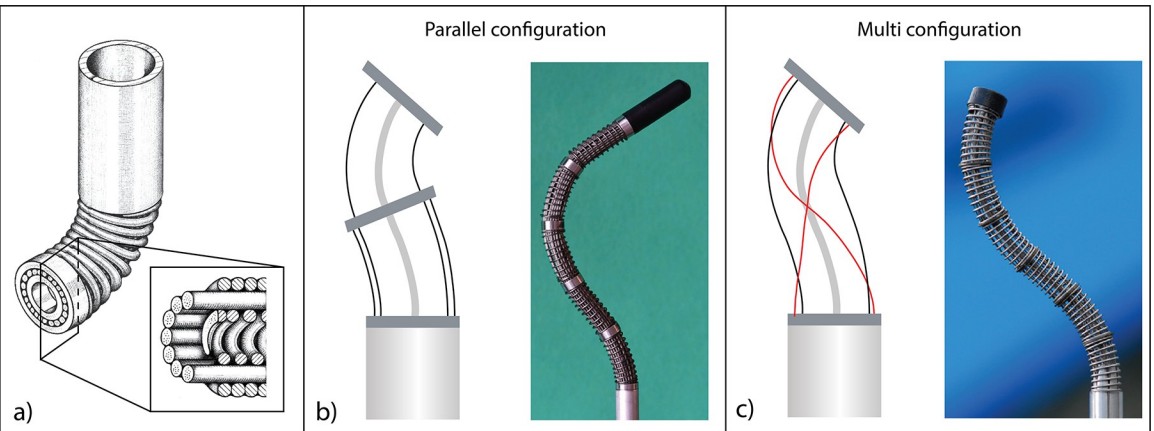

**Fig 1. Multi-steerable strategies to control surgical instruments.** a) Cable-ring mechanism with its cross-section. Cables are placed concentrically to actuate the segments and guide each other along the shaft, adapted from [23]. b) Parallel configuration of a multi-steerable instrument, adapted from [25]. c) Multi configuration of a multi-steerable instrument, adapted from [17].

steeper learning curve, faster task performance time, requires a lighter workload, and is preferred by the participants.

## 2. Cable configuration strategies

Cable-driven steerable instruments are controlled by actuation forces applied to their steering cables. In multi-steerable instruments, various deformation modes can be generated with different cable configurations, determining the behavior of the steerable shaft. In a 2D representation of one segment, we can define a generic steerable segment as an incompressible compliant backbone, with a length L, in which a rigid end plate of 2R in length is attached at the distal end, Fig 2. The proximal end of the backbone is fixed and represents the connection with the shaft. Actuation cables are attached at the outer ends of the end plate. In the case of a 2D symmetrical cable configuration, each segment can have two parallel cables (the parallel configuration), two diagonal cables, or the combination of diagonal and parallel cables (the multi configuration).

### 2.1 Parallel configuration

In the parallel configuration, cables are placed parallel to the backbone, and the pulling force $F_p$ is parallel to the longitudinal axis of the segment, Fig 2A. Therefore, the bending moment is constant along the segment length L because the perpendicular distance R between the force application point and the backbone stays constant along the segment length L. The bending moment, therefore, defines the orientation angle of the segment (β), and the deflection mode will result in a curve with a constant bending radius. Segments with a parallel configuration can be combined by placing them on top of one another so that the base of the first segment acts as the top of the second segment and so on. The combination of the segment angles defines the position and the orientation of the end-effector, i.e., the end plate of the most distal segment, allowing different deformation modes.

### 2.2 Multi configuration

In the diagonal configuration (Fig 2B), cables connect the end plate to the fixed base by crossing each other forming the α angle fo Fig 2B. Forces $F_d$ are applied along the cable direction

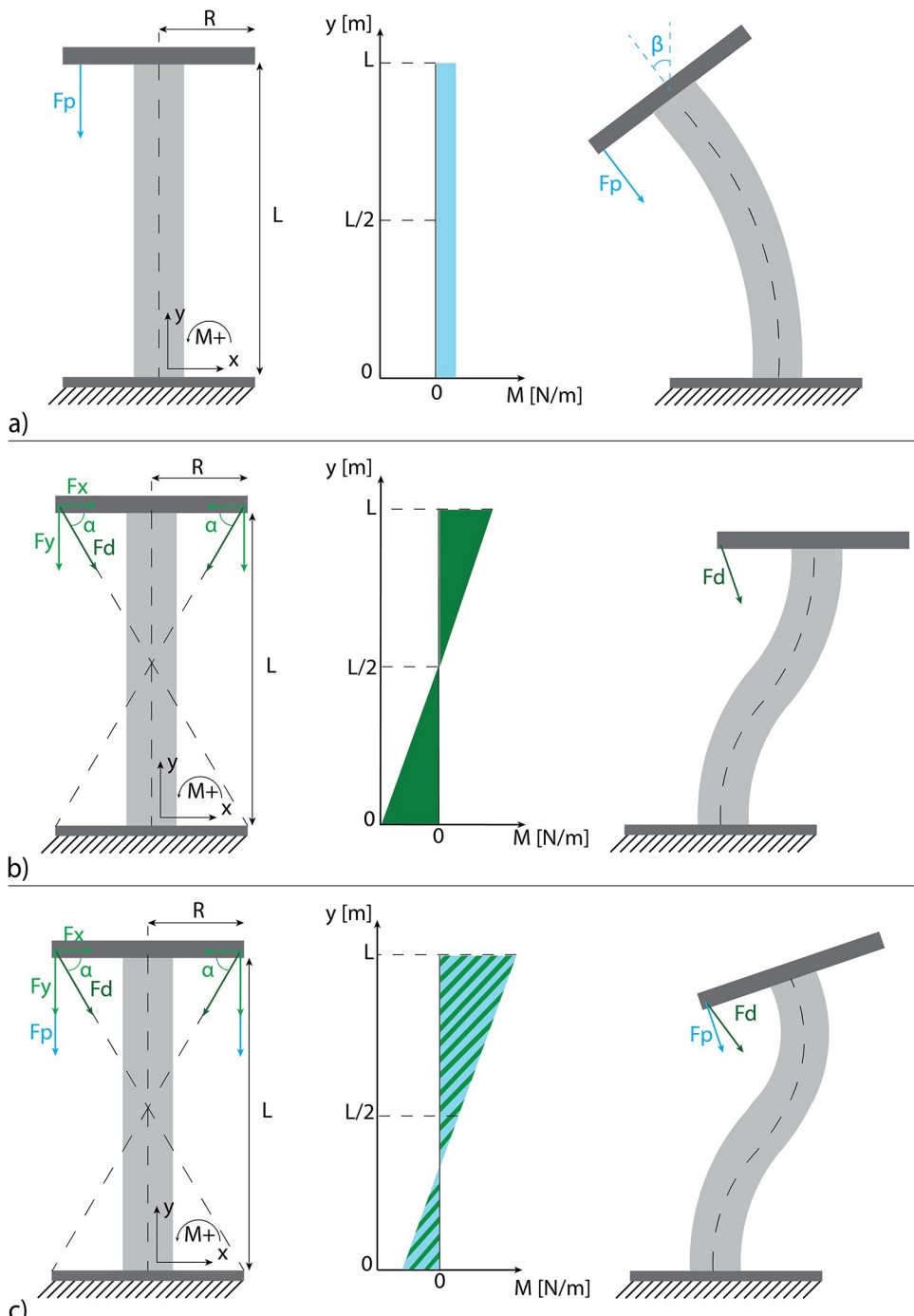

**Fig 2. The three cable configurations presented in this work.** Left: 2D representation of the segment, center: Corresponding bending moment diagram, right: Segment deforming under the applied pulling force. a) Parallel configuration, b) diagonal configuration, c) multi configuration. For the explanation of the used symbols, see the text.

and can be split into the $F_x$ and $F_y$ components. When the cables cross each other at L/2, the segment will have a symmetric bending moment and, therefore, symmetric behavior, enabling a double-curved shape deformation mode as shown in Fig 2B. In this case, the end plate

translates laterally in the direction of the $F_x$ force while the orientation of the end plate remains unchanged.

Full control with only one segment can be obtained by combining parallel and diagonal cables, Fig 2C. This combination, which will be referred to as the multi configuration [17], results in a mechanical behavior similar to the parallel configuration but with only one segment instead of multiple segments.

## 2.3 Three-dimensional representation

Navigation through confined anatomy requires instruments able to move in a 3D space. For instruments based on the *parallel configuration*, 3D motion can be achieved by using a minimum of three actuation cables per steerable segment. However, the use of four cables per segment concentrically placed at a 90 degrees angle allows antagonist movement of the cables and simplifies control [24]. The steerable segments are placed in series, one after the other, to increase the DOF of the shaft. The combination of a number of segments allows the control of the orientation as well as the position of the end-effector. In an instrument with multiple steerable segments, the actuation cables that control the end-effector run through dedicated slots of the preceding segments, the cables of the first preceding segment through the preceding ones, and so for all segments, Fig 3A. To avoid overlap of the actuation cables, each steerable segment of the shaft is rotated slightly, as shown in the close-up of Fig 4A.

For instruments based on the *multi configuration*, 3D motion can be achieved by placing the parallel cables concentrically at a 90 degrees angle, similar to the parallel configuration but diagonal cables need a reconfiguration, Fig 3B. In fact, the diagonal cables will cross the backbone if positioned like in the 2D representation, and they will not allow any internal lumen to be included in the instrument. A possible solution, which was successfully investigated by Gerboni *et al.* [17], is to use helically-oriented cables that rotate 180 degrees around the central backbone [26]. Rotation of the helical cables can be either in the clockwise or in the counterclockwise direction. Using only one of the two directions would lead to an undesired torque along the segment backbone. Combining clockwise and counterclockwise helical cables in pairs cancels out this effect. In the parallel configuration, the parallel cables of different segments can be placed at the same distance from the backbone due to the straight nature of the cable slots. In the multi configuration, the helical cables would cross each other, causing overlapping of the cable slots. In order to avoid this arrangement, the three sets of cables (clockwise, counterclockwise, and parallel) are placed concentrically at three different radii as close as possible to each other, but still remaining independent by dedicated grooves Fig 4B. Differently from the parallel configuration in which we need multiple segments to determine the position of the end-effector, in the multi configuration, we can consider the steerable shaft as one long steerable segment due to the possibility of controlling both the position and orientation of the end effector with four actuation cables of each type (clockwise, counterclockwise, and parallel).

## 3. Instrument prototypes

### 3.1. Design

Two prototypes were designed with an identical outer appearance and size: one based on the parallel configuration and one on the multi configuration. Both prototypes contain three components: a compliant handle, a rigid shaft, and a compliant shaft. A detailed description of the compliant shaft and the design in the parallel configuration is described by Culmone et al. [24]. Both designs are based on a cable-driven actuation with a serial control strategy, in which the movements of the compliant handle and the one of the shaft are mirrored [21]. The

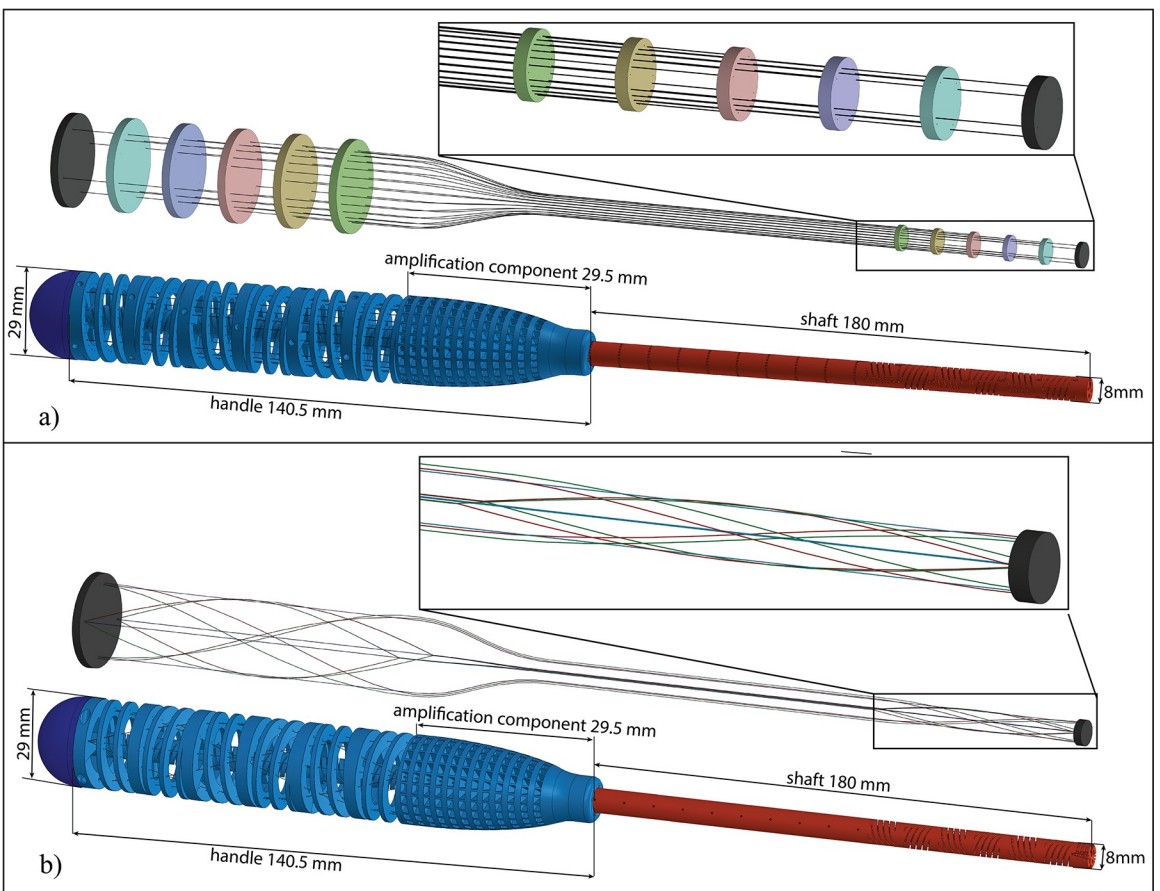

**Fig 3. 3D model of the parallel and multi configurations.** a) The parallel cable configuration with a close-up on the shaft and the final design of the device. Each color corresponds to a segment mirrored on the control side. b) Multi configuration of the cables and the final design of the device. In this configuration, all cables are connected at the ends. The close-up shows the cable configuration in the steerable shaft.

compliant shaft is based on a modular compliant segment, composed of a central flexible back-bone and four helicoids that run concentrically around the centerline, Fig 5. The helicoids have a T-shaped cross-section that is thin close to the backbone to ensure low bending stiffness and enlarges towards the outer side of the segment to limit the bending angle and prevent failure for excessive bending, Fig 5B. Segments with helicoids inversely placed around the back-bone (clockwise and counterclockwise helicoids) are alternately placed on top of one another to form the compliant shaft and ensure equally divided torsion stiffness around the backbone. The actuation cables run through holes in the helicoids and are looped into a cross-shaped groove at the top of the steering segment to fix and control them independently, Fig 5C and 5D.

The rigid shaft is connected directly to the compliant shaft. The rigid shaft contains dedicated slots to guide the actuation cables from the compliant shaft to the handle. The compliant handle is based on using wrist control, in which all fingers and the wrist are used to manipulate the handle to define the desired shape of the compliant shaft. The design of the compliant handle is similar to a large version of the compliant shaft. For each compliant segment of the shaft, there is a respective segment for the handle. Similar to the compliant shaft, each segment of the handle has an inner backbone surrounded by an outer helicoid structure. The helicoid structure has the function of guiding the cables as well as creating a cable fixation point. The

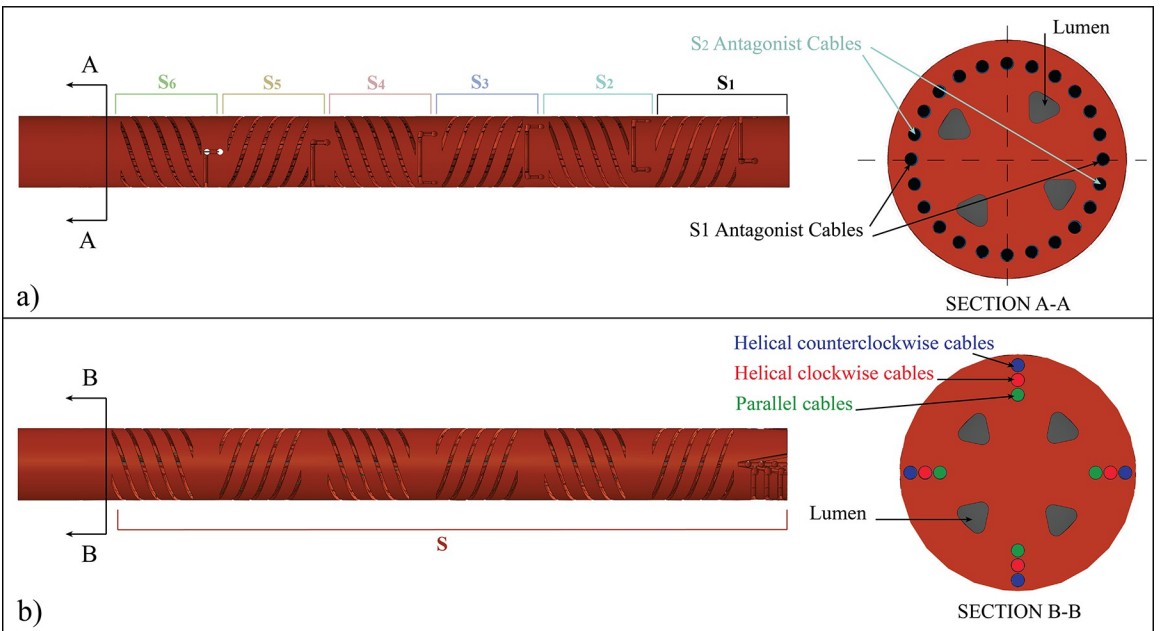

**Fig 4. Cross-section of the shaft for the parallel and the multi configurations.** a) In the parallel configuration all cables are equally distant from the central backbone Segments *S* are numbered from 1 to 6. In the multi configuration, cables are concentrically placed at three different radii to avoid overlapping. *S* represents the steerable shaft.

connection between the rigid shaft and the handle is smoothened by an amplification component, the rigid distal part of the handle that, with an amplification factor of three, guides the cables from the shaft to the handle, amplifying the movement between the handle and the shaft (Fig 3). Moreover, both designs have four lumens to insert flexible thin tools for diagnostic or treatment.

In the two prototypes, the steerable shaft shares the same design based on six steerable segments. However, in the parallel configuration, actuation cables control every single segment

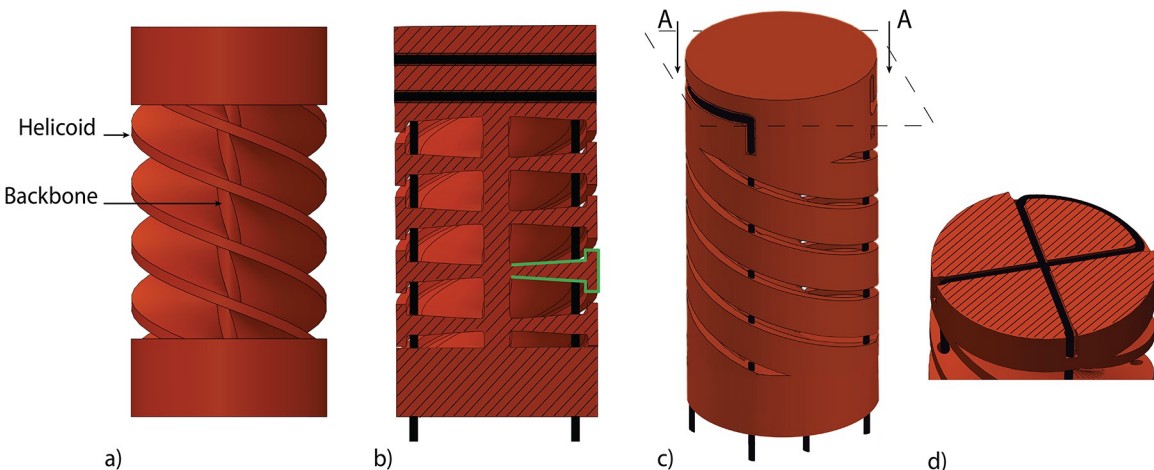

**Fig 5. Steerable segment with parallel cables.** A) Four helicoids concentrically placed around the central backbone. b) Cross-Section of the steerable segments with the T-shape of the helicoids highlighted in green and cables in black. c) 3D model of the steerable segment with d) cross-section A-A showing the looped cables in the fixation point. Adapted from [24].

independently, and the cable fixation point needs to be located directly on the segment itself, in both the compliant shaft and the handle. In the multi configuration, the six segments are considered as one element. All actuation cables run through the entire body of the instrument and are fixed at its two ends: at the distal end, the end-effector, and at the proximal end, the end of the handle. Therefore, while in the parallel configuration, the position and orientation of the end-effector are controlled indirectly by controlling the orientation of the individual segments, in the multi configuration, the position and orientation of the end-effector are directly controlled as if there is only one segment. Moreover, different from the parallel configuration where cables run through straight guiding slots, in the multi configuration, the guiding slots for the cables are both on straight and helical tracks. The parallel configuration uses a serial control strategy based on mirrored movements. The segments of the shaft are mirrored in the handle, Fig 3A, resulting in the shaft moving opposite to the handle, i.e., the end of the handle moving upwards resulting in the end-effector moving downwards. Also, the multi configuration is controlled by mirrored movements: when a cable is pulled by bending the handle, it will shorten in its distal end, mirroring the handle movement. With the equal length in the steerable shaft, the two cable configurations are able to cover the same workspace and perform single curved and double curved shapes. Moreover, having an identical appearance for the two prototypes allows for comparing the cable configuration performance without influencing the participants at the test.

## 3.2. Fabrication

Both instrument prototypes were fabricated using vat photopolymerization as additive manufacturing technology, Fig 6. All parts were printed using Perfactory® Mini XL (EnvisionTEC GmbH, Gladbeck, Germany), with 25 μm layer height in the vertical z-axis. The printer, based on the so-called Digital Light Processing (DLP), uses a light source and a projector to harden the liquid resin layer by layer. We used R5 (EnvisionTEC GmbH, Gladbeck,

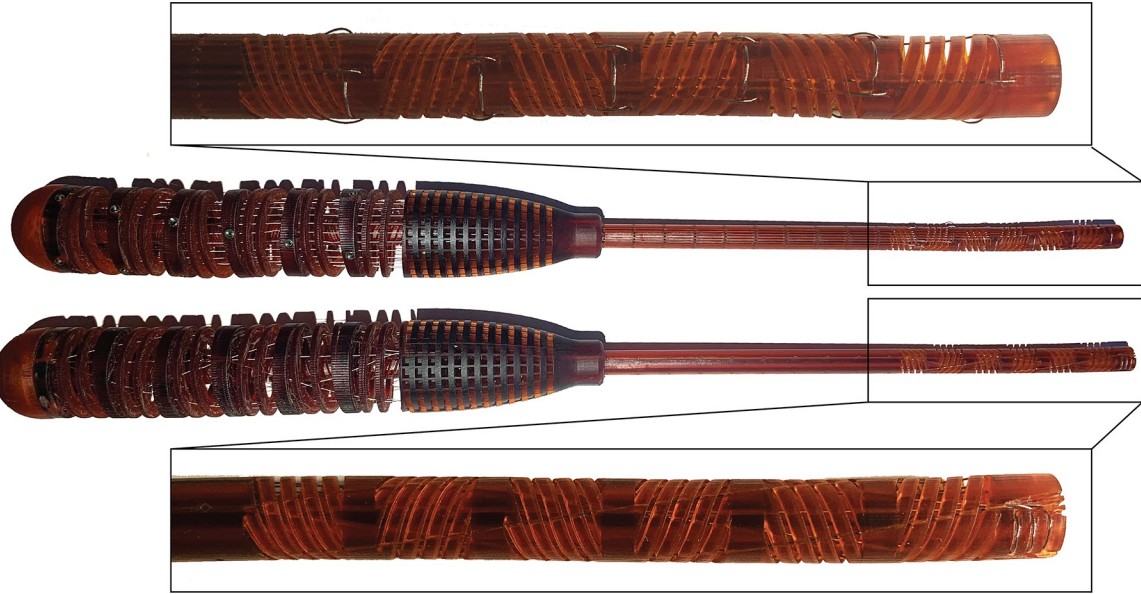

**Fig 6.** Instrument prototypes employing the parallel configuration (top) and multi configuration (bottom). The close-ups show the steerable shafts. Notice that the fixation points in the shafts differ, depending on the configuration. In the parallel configuration, each segment has two fixation points whereas in the multi configuration all cables are fixed at the distal end of the shaft.

Germany), an epoxy photopolymer resin, which is specifically customized for prototyping. The handle and the shaft were printed in the vertical position, with the main axis parallel to the z-axis of the printer. The shaft was printed without any support except for the raft, a layer between the built plate and the printed object. The handle was printed with a raft, and an internal support made of small pillars autogenerated by the printer software. After printing, the excess resin was removed by placing the parts in an isopropyl alcohol bath for 30 minutes. The raft and the support of the handle were manually removed. The dimensions of the printed devices slightly differ from the CAD model with a tolerance of ± 0.08mm for the shaft and ± 0.12 mm for the handle. A total of 24 ∅ 0.2 mm stainless-steel cables in the parallel configuration and 12 ∅ 0.2 mm stainless-steel cables in the multi configuration, were used to actuate the instruments. To avoid overlapping, the radii of the cable circle in the multi configuration were determined considering the printability of the instrument and the cable size. Once decided the amplification factor by fixing the diameter of the handle, the cables were placed as close as possible to each other but at the same time kept independent with dedicated grooves. Moreover, the order of the cables (clockwise, counterclockwise, and parallel) was decided considering the easiest way to mount the cables into the instrument. After placing them in the 3D printed construction, the cables were straightened using weights of 3 grams and then fixed in the handle by means of dog point set screws. At the end of the assembly, a small steel plate was glued at the end-effectors for use in the experiments.

## 4. Functionality evaluation

### 4.1. Background and goal

A comparative evaluation was carried out to study the maneuverability of the instruments and investigate which cable configuration enables a faster and easier control strategy. We hypothesized that:

- The parallel configuration requires less workload. The straight arrangement of the cables within the instrument generates less friction due to the lower normal forces between the actuation cables and the 3D helical printed structure as compared to the multi configuration.

- The multi configuration would be faster in hitting the target, considering that all cables control the entire steerable shaft at once, whereas, in the parallel configuration, each steerable segment of the handle individually controls the corresponding steerable segment of the shaft.

The two instruments were tested in a laparoscopic simulator where targets with different orientations and positions were placed. The participants had to hit the indicated targets as fast as possible. The task was repeated 12 times (runs) per trial. Each participant attended four trials, two for each instrument. For each run, we measured the time to complete the task properly. We analyzed and compared the task performance time between the two instruments. Moreover, we examined the learning curve for each instrument and whether the order of use influences the learning curve. Finally, we analyzed the experienced workload and the individual preference of the participants using questionnaires.

### 4.2. Participants

Based on similar studies [13,27–29], a total of 12 participants (5 men and 7 women, aged 27.4 ±1.9) were recruited to take part in the experiment. All participants had no prior experience in laparoscopic or open surgery procedures, nor with laparoscopic instruments and were recruited within the BioMechanical department of Delft University of Technology (master

students, PhDs, and technicians). Participants were all right handed with different videogames habits, between 0 to 10 hours a week. One participant played a musical instrument. The participants were split randomly into two groups, Group A and Group B, of 6 participants each. Each group had a different order of instrument use. Group A started with the instrument with the parallel configuration (PC), whereas Group B started with the instrument with multi configuration (MC). All participants were informed about the purpose, the type of the experiment, and the use of the collected data. The study was approved by the Human Research Ethics Committee at Delft University of Technology (ID:1408).

## 4.3. Experimental setup

The experiment was carried out using a laparoscopic simulator, specifically designed for this study. The simulator was designed to create, the movement that the surgeon might perform while navigating a laparoscopic procedure, in a simulated environment. Particular importance was given to the precision and orientation of the instrument, essential for preserving the surrounding critical areas. The simulator was made of clear PolyMethylMethAcrylate (PMMA) and PolyPropylene (PP) to replicate an inflated abdomen. We decided to have a transparent simulator to provide participants with direct 3D visualization. Due to their inexperience in laparoscopic surgery, using an endoscope and a monitor could have resulted in additional difficulties related to the loss of depth rather than the instrument maneuverability. Using 2D visualization of the target area, the collected data would not reflect the learning curve of the participants to properly operate the instruments, but rather their learning curve in visualizing the space in 3D from a 2D image. A silicon valve placed in the center of the simulator allowed the insertion of the instruments. A 3D printed cylindrical stand with seven targets in different orientations was placed inside the simulator. Six target tubes (20 mm long, ∅ 9 mm) were numbered and evenly placed around the stand, while a start flat target was placed center of the stand. The entrance point of the target tubes was marked with a black line. Each target tube contained two steel plates at its bottom, Fig 7E. When the end-effector was parallel oriented to the plates and therefore hit them simultaneously, electric contact was made, and a signal was measured by a Multifunction I/O device (USB-6008, National Instruments, Austin, USA) [30] that was controlled with a laptop via a LabView 2016 program (National Instruments, Austin, USA). A monitor showed the next target to be hit. The setup and the monitor were positioned in front of the participant, and their height could be adjusted to reach a comfortable position, Fig 7.

## 4.4. Task and procedure

The task consisted of positioning and orienting the multi-steerable shaft to reach the six targets. The experiment started when the participant hit the start target. Subsequently, the participant was asked to move the shaft towards the indicated target (randomly chosen among the six) and insert the tip into the tube. A low-frequency buzzer indicated that the participant hit the two steel plates of a wrong target, whereas a high-frequency buzzer indicated that the correct target was hit and the participant could move the shaft towards the new target. The time was recorded and was measured from the moment the participant hit the start target until the last target was hit. In each run, the participant hit the start target and the six other targets in a randomized order. Each trial consisted of 12 randomized runs. Each participant performed four trials: two trials for each of the two cable configuration instruments, allowing a comparison between the two groups over the two configurations and resulting in a total of 48 runs (12 runs x 2 trials x 2 cable configurations) per participant. The number of attempts per trial has been based on previous works where similar tests for steerable instruments have been

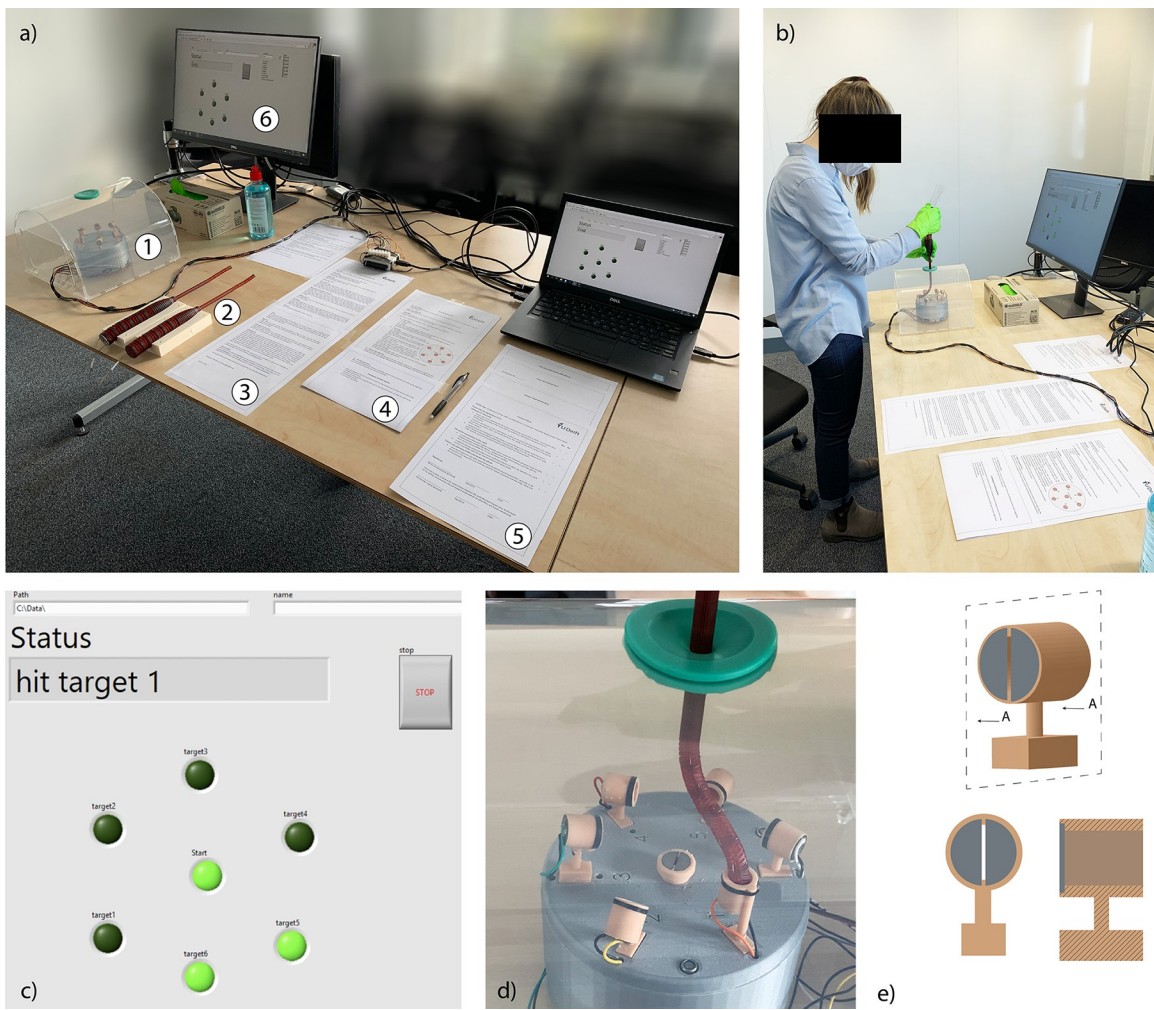

**Fig 7. Experimental setup.** a) Setup and its components: 1. simulator, 2. instruments, 3. participant information letter, 4. general instruction, 5. informed consent and questionnaires, 6. user interface. b) A participant during the test. c) The user's interface during the experiment. Each green circle represents a target. The light green circles are the targets already hit, and the dark green circles are the targets still to be hit. The status bar displays the next target that the participant has to hit. d) The instrument into one of the targets. d) A participant during the test. e) CAD model of the target with back view and cross-section. The two steel plates are represented in grey.

performed [13]. Prior to the start, a short demonstration and an instruction sheet were given to the participant. The participant filled up an intake questionnaire with general information such as gender, age, educational phase, dominant hand, and video game or musical instruments experience. Before each of the four trials, the participants had two minutes to practice and familiarize themselves with the instrument. For participants of Group A, the experiment sequence was PC instrument followed by MC instrument, and again PC and MC. For Group B, the experiment sequence was MC-PC-MC-PC. In the supplementary materials, S1 Fig. shows the flow chart of the experiment and the two instruments order for the two groups.

At the end of each trial, the participant had a break of around 10 minutes to fill a self-evaluation questionnaire based on NASA's Task Load Index (TLX) [31]. The six subscales (mental demand, physical demand, temporal demand, performance, effort, and frustration) of NASA TLX were rated from -10 to 10, in which a high score indicated that the task was highly demanding and a low score that was easy to perform. At the end of the fourth trial, the

participant filled out a final questionnaire to express a preference between the two instruments, considering the ease of steering and control. All data were analyzed using Matlab R2020a scripts (accessible in the data availability). The S1 Video in the supplementary material shows the execution of one run for each instrument.

## 5. Results

Fig 8 shows the task performance time per instrument in the two trials. Yellow represents the PC, and blue the MC. The plot depicts the results as box and whiskers, where the bottom edge of the box indicates the 25th percentile, the top edge the 75th percentile, and the red central line the median. The median time for trial one was 102.05 s for the PC and 106.60 s for the MC. In Trial 2, the median time was 74.15 s and 76.75 s for both configurations, respectively. The median decreased for both instruments between the first and the second trial. Due to the asymmetry of the data calculated with the Shapiro-Wilk test (p<0.001), we performed the Mann-Whitney U test for independent groups of non-parametric data. The test revealed no significant difference (Z = -0.72, p = 0.44>0.05) on the task performance time of the two devices in each trial. Moreover, we compared the two trials of the same cable configurations. In both cases, the Wilcoxon Signed-Rank test for two dependent groups of non-parametric data showed a significant difference between the two trials (Z = 8.32, p<0.05), and therefore a

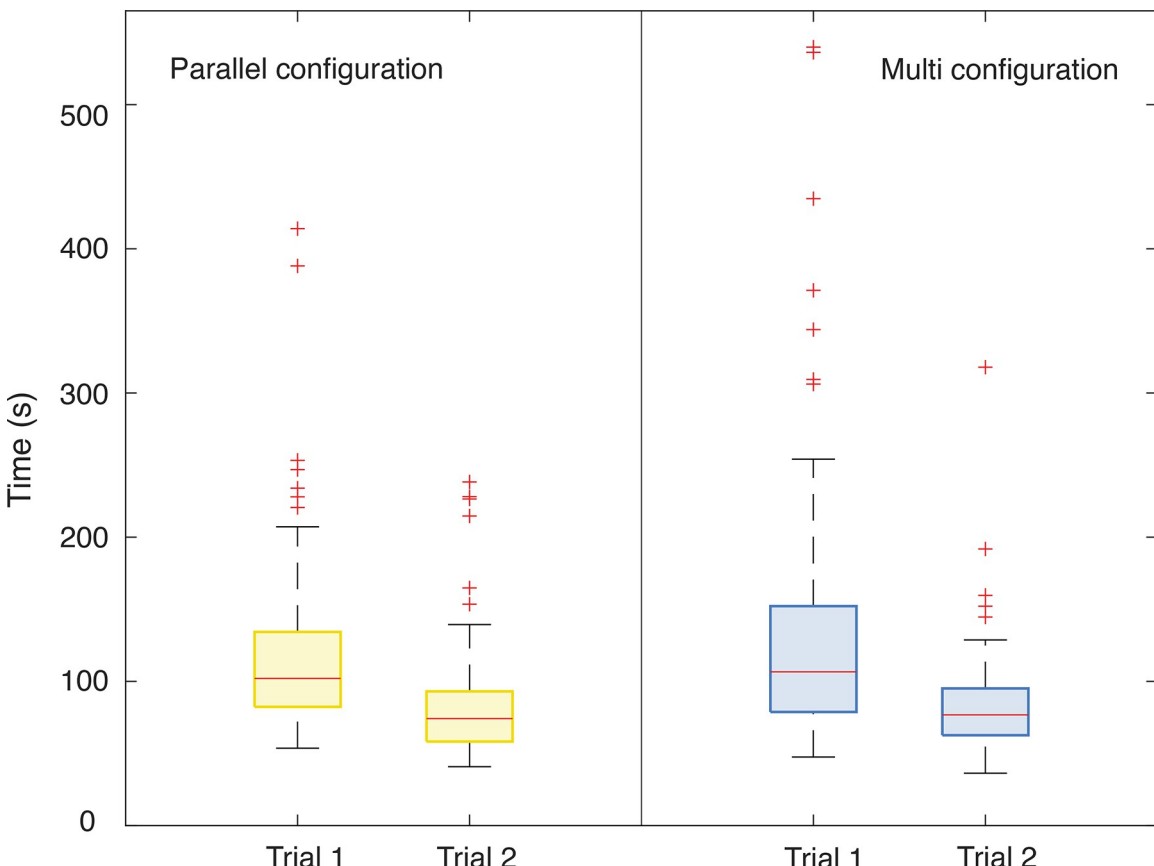

**Fig 8. Box and whisker plots of the task performance time for the two cable configurations.** Yellow represents the parallel configuration (PC), and blue the multi configuration (MC). For each instrument, the participants performed two trials. The red line in the box represents the median and the red crosses, the outliers.

significant reduction in time between the first and the second trial for both configurations as the participants got more experienced with the instruments after some training.

Looking at the trend of the runs within the trials, Fig 9 shows the learning curve of the participants per each instrument within the 12 performed runs of each of the four trials. The average time shows a reduction of 48% for the PC and 41% for the MC calculated as the difference of the average time between the first and the last run of the first trial. Data stabilized in the second trial for both instruments with an average time reduction of 24% for the parallel and 14% for the MC. The time performance for the PC and the MC in the last run of the second trial shows similar results: 76.42±19.87 s for the PC and 74.99±21.99s for the MC.

The minimum task performance time average was 49.92±8.92 s and was achieved by 9 out of 12 participants in the last performed trial, Trial 2 with the MC for Group A, and Trial 2 with the PC for group B. Two participants of Group B achieved the minimum task performance using the MC; one participant in Trial 1 and one in Trial 2. In Group A, one participant achieved the minimum task performance time in Trial 2 with the PC.

The maximum task performance time, with an average of 272.15±124.85 s, was achieved in the first performed trial for 11 out of 12 participants, independently from the instrument. Only one participant of Group A achieved the maximum task performance time in Trial 1 with the MC.

Moreover, we looked at the influence of one instrument over the other, considering their order. Fig 10 shows the box and whisker plots of the task performance time for the 12 participants in the four trials for each run. We compared the task performance time of the first run of the two groups, A and B, in Trials 1 and 2 for the PC and MC, Fig 10. We performed the Mann-Whitney U test for independent groups of non-parametric data. The test revealed a significant difference (Z = 2.85, p<0.05) between Group A and Group B in Trial 1 for the PC. Group A (which started with the PC) required more time with an average time of 254.60 ±119.12 s than Group B (which started with the MC), which required 113.98±9.65 s for the same task in Trial 1. Also, for the MC there was a significant difference (Z = 2.43, p<0.05) between both groups in Trial 1. Group A required 124.30±13.98 s, which is less than Group B, which required 210.47± 84.53 s. Among the four trials, the learning curve of the participants shows a decrease in average task performance time. The curve dropped by more than 55% in

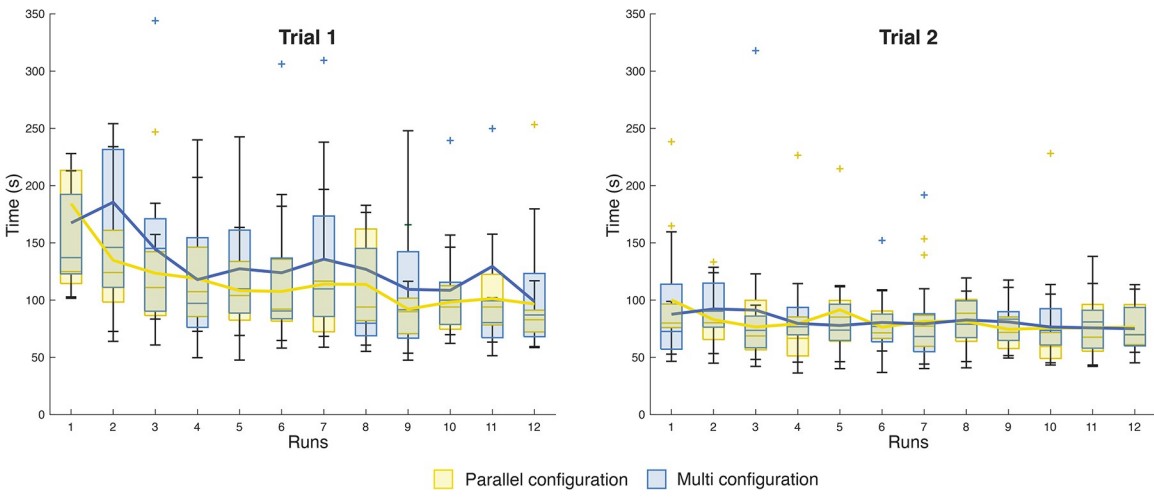

**Fig 9. Box and whisker plots of the average time per run in the two trials performed by each participant for each instrument.** Yellow represents the parallel configuration (PC), and blue the multi configuration (MC). Each box and whisker plot represents the median, the upper and the bottom quartile of the average time for 12 participants. The outliers above 350 s have been cut off in the figure. The full picture can be found in the supplementary material.

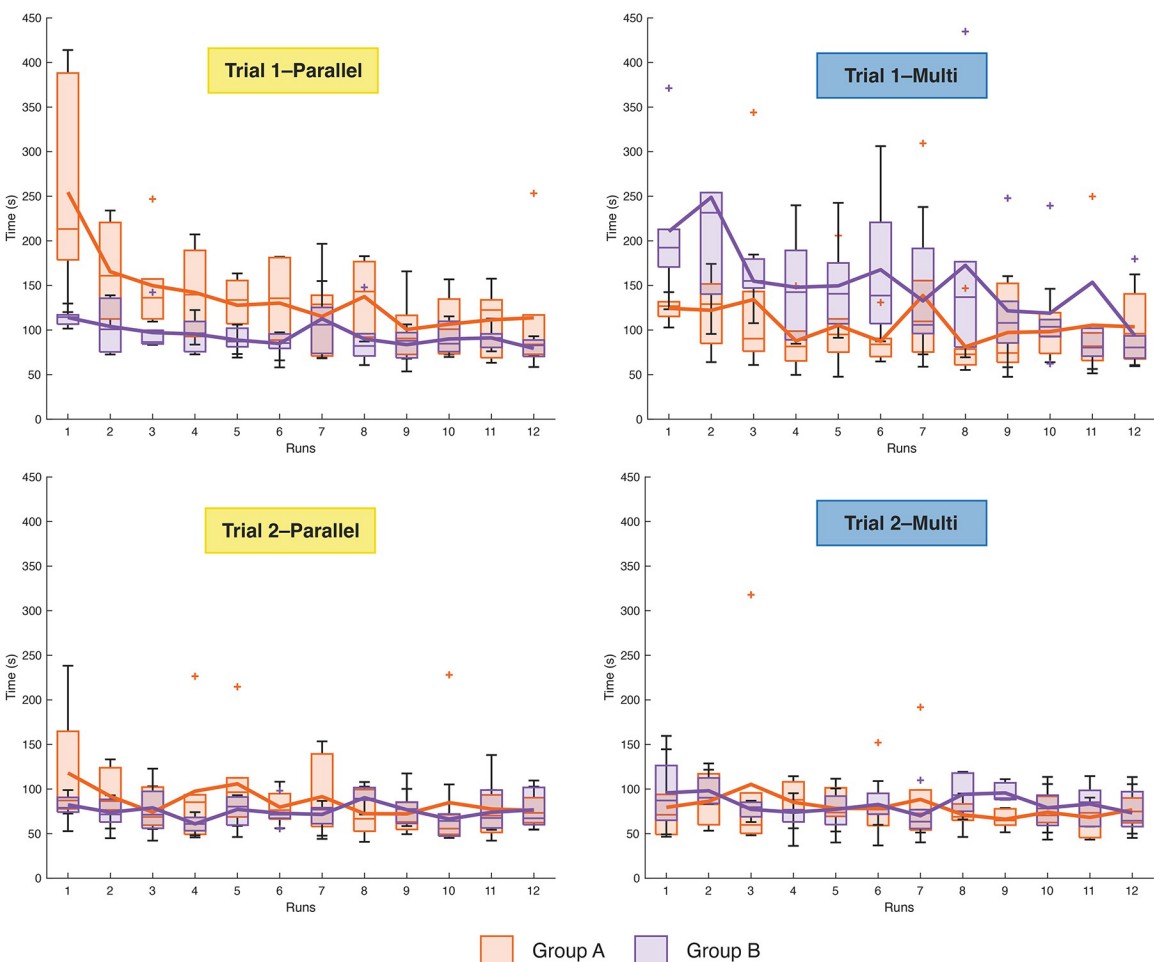

**Fig 10. Box and whisker plots of the average time per run in the two trials performed by each participant for each instrument in the two different groups.** Orange represents Group A and purple Group B. Each box and whisker plot represents the median, the upper, and the bottom quartile of the average time for the six participants of Group A and the six of Group B.

task performance time between the average time of the first and the last run performed by all the 12 participants with ~~for~~ the very first instrument used, no matter which configuration, and flattened to a decrease of 3–7% for the very last instrument used in both groups.

The responses of the TLX self-evaluation that ranged from -10 to 10 were transferred to a percentage scale. High percentages express a high workload, and low percentages express a low workload, i.e., -10 expresses 0% workload, whereas 10 expresses 100% workload. The overall Raw TLX score was 34% (SD = 22) for the parallel and 40% (SD = 23) for the MC in Trial 1. In Trial 2, the overall Raw TLX score was 23% (SD = 22) and 30% (SD = 23) for the parallel and the MC, respectively, Fig 11. We performed the Mann Whitney U test for independent groups of non-parametric data. The test revealed no significant difference (Z = -1.55 p = 0.12>0.05) between the overall workload in the first trials of the two instruments. In the second trial, the overall workload was significantly higher (Z = -2.18, p<0.05) for the MC compared to the PC. The Wilcoxon Signed-Rank test revealed a significant reduction (Zp = 4.92 Zm = 4.58, p<0.05) in the overall workload of the two instruments from Trial 1 to Trial 2. Participants expressed the maximum workload for both instruments in the effort subscale of Trial 1, 49% (SD = 21) for the PC and 57% (SD = 25) for the MC, respectively.

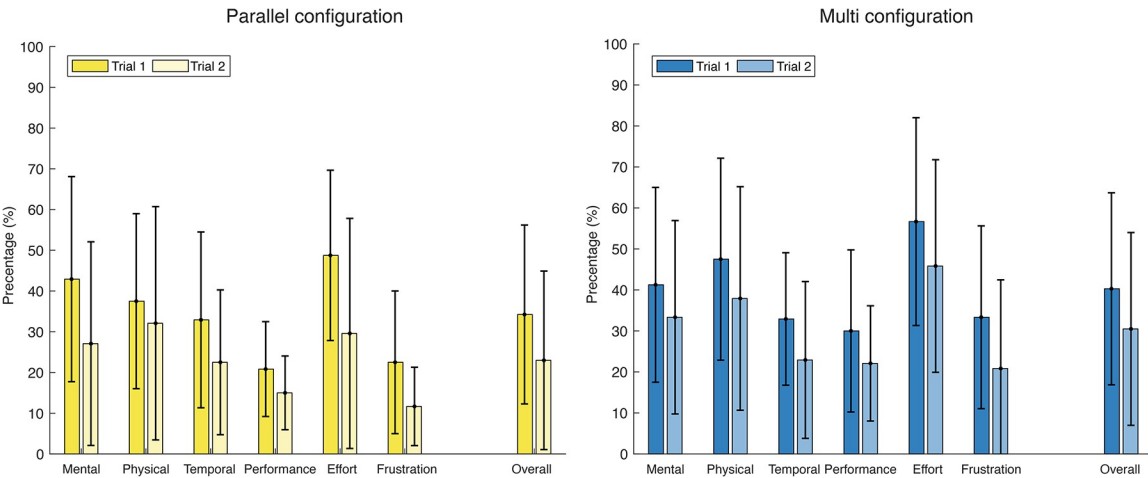

**Fig 11. Average and standard deviation of the Raw TLX score for the six subscales (mental demand, physical demand, temporal demand, performance, effort, and frustration) in Trials 1 and 2.** The average was calculated over the score given by the 12 participants. Yellow represents the parallel, and blue the multi configuration.

Finally, Fig 12 shows the result of the final questionnaire on the subjective participant preference. The participants expressed a strong overall preference for the PC, 10 out of 12. All participants preferred the PC when considering the response in steering.

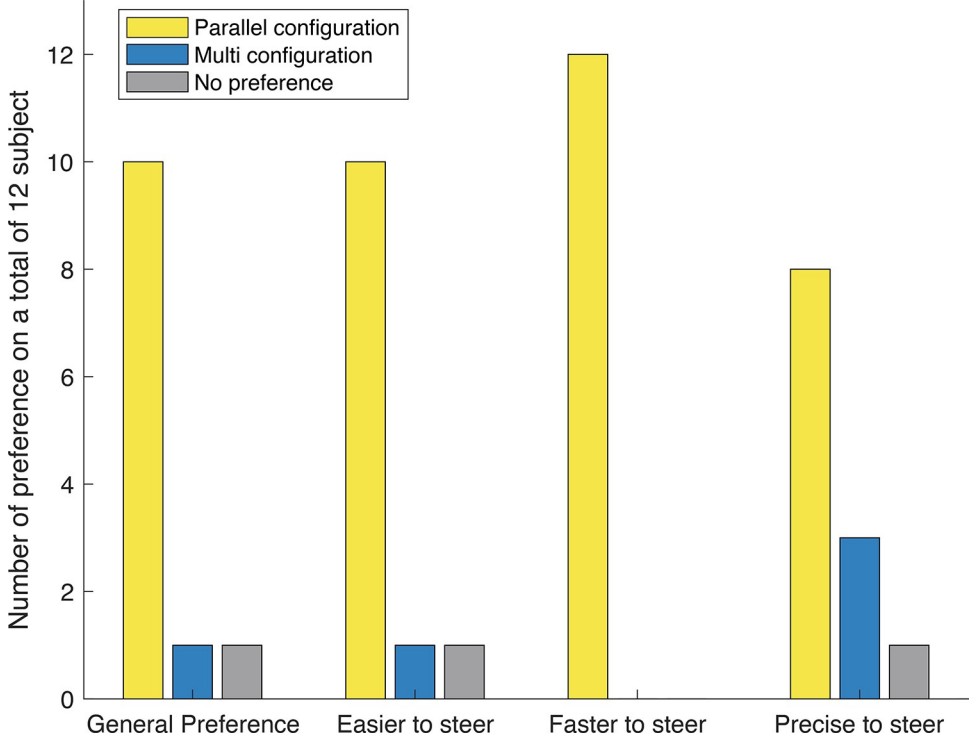

**Fig 12. Results of the final questionnaire on personal preference.**

## 6. Discussion

### 6.1. Experimental findings

The difference in task performance time was not significant when comparing the parallel and multi configurations to each other over Trials 1 and 2. A significant decrease appeared over time within the two trials when using the same configuration. This data was also confirmed by the learning curve of the two configurations. The two learning curves showed that the task performance time decreased quickly in Trial 1 during the first runs, and the participants reached a plateau after the first runs of Trial 2 for both instruments. It is interesting that for the parallel configuration, the minimum average task performance time was reached during Run 9 of Trial 2, instead of the last run, with a slight increase in average time for the subsequent runs. This effect is probably due to the tiredness of the participants at the end of the test. The flattening of the curves also showed its effect on the decrease in the workload perceived by the participants. The time performance for the parallel and the multi configurations in the last run of the second trial shows similar results for the parallel and the multi configuration, rejecting our second hypothesis.

The workload strongly decreased from Trial 1 to Trial 2 for both configurations. However, even though the task performance time did not show significant differences between the parallel and the multi configuration, the decrease in workload was significantly higher for the parallel configuration. This result can also explain the net difference in the preference of the parallel configuration over the multi configuration.

Looking at the alternation between the two cable configurations over the four trials, it becomes clear that the instruments influence each other over the first trials. In Trial 1, Group A started with the parallel configuration, and the average task performance time is significantly higher than the one in Trial 1 of Group B (which started with the multi configuration) for the same configuration. The same result can be observed for the opposite: the task performance time of Group B with the multi configuration in Trial 1 is significantly higher than the one of Group A. In the very first run, when they used their first instrument for the first time, the participants needed not only to learn to use the instrument and gain dexterity but also needed to familiarize themselves with the setup and the target positions. When they used their second instrument for the first time, they only needed to get used to the different cable configurations.

We also analyzed the performance of the participants within each run. An interesting outcome was the target that required the longest time to be hit and its occurrence within all 48 runs. The analysis revealed that Target 6 was the most difficult target to be hit, 195 times out of the total of 576 recorded runs. This can be explained by the location of Target 6, which was located the closest to the participant, requiring the instrument tip to be pointed towards the participant, mirroring its motion and thus adding an extra layer of difficulty in the maneuverability. The analysis becomes even more interesting when Target 6 is compared to Target 4. Target 4 has the same orientation angle but has an opposite location of Target 6. Target 4 recorded only 66 times the longest time to hit the target, the lowest occurrence among all targets. This result is probably due to its convenient location at the front right of the participant.

The net preference of the parallel over the multi configuration was briefly explained by four participants in the comments at the end of the final questionnaire and was mainly related to steering possibility. The parallel configuration gives the possibility of steering the segments independently, which is especially convenient for the most distal segment. By individually controlling the most distal segment, the participants felt more control over the final shaft orientation during insertion into the target. Therefore, the parallel configuration showed easier maneuverability over the multi configuration as hypnotized. On the other side, the multi configuration was noticed to be faster in reaching the initial position to hit the target, but less

precise when aligned the tip to the target. Moreover, the multi configuration was preferred for the higher stiffness of the entire instrument, which allows for stronger haptic feedback of the steerable shaft during the test. The higher stiffness perceived by the participants was probably caused by the friction generated by the higher normal forces between the tensioned helical cables and the 3D printed helical structure in the handle as compared to the parallel cables. This observation is also interesting considering that the total number of cables in the multi configuration is half of the one in the parallel configuration. The different characteristics in speed, stiffness, and precision of the two cable configurations might open two different paths for the instruments. Whereas the multi configuration is used when the speed of the task is the main challenge, the parallel configuration is used when precision is fundamental to success in the task. It is also important to notice that all participants completed the test and no significant increase in the performance time was recorded for any of them at the end of the test. This consideration is important for the evaluation of the instrument maneuverability in view of possible future studies.

## 6.2. Limitation of this study and future recommendation

Additive manufacturing (AM) represents a significant innovation in terms of fast prototyping and the complexity of the design. In our work, AM allowed us to print highly complex compliant structures enabling advanced instrument maneuverability with very limited assembly time —the instruments were printed and assembled in less than one and a half days. Our study was mainly focused on device maneuverability and functionality. Therefore, the instruments were fabricated with an acrylic-based polymeric resin, which was non-biocompatible but specifically designed for easy and precise prototyping. Future work should focus on investigating the use of biocompatible materials able to guarantee the same compliant characteristics of the material used in this study. We think that our instruments should, in the end, be used as disposable devices, opening possibilities for the patient and surgeon-specific designs.

We used the same instrument for more than one participant, and, to always have fully functional instruments, in our experiment, we decided to use a new instrument every time we noticed signs of failing. All participants used the same two devices from the beginning to the end of the test (for all the 48 attempts) except for one participant for whom the multi configuration device needed to be replaced due to breakages on the end-effector side. Most of the time, the breakages were associated with excessive force applied by the participant to hit the target, and they were mainly on the end-effector side. Another reason for failure was due to the wear of the polymeric-based material induced by the stainless-steel cables again on the end-effector side. The stainless steel cables showed no signs of fatigue when straightened, however, especially during the first few attempts when the participant familiarized with the tool, excessive bending of the handle resulted in local bends.

The test was performed under direct 3D vision due to the inexperience of the participants with laparoscopic procedures. This choice was made as no experienced particpants were available, due to unforeseen limitations due to the global pandemic. Performing the test with an endoscope and a monitor would improve the resemblance of the task with the clinical setting. Moreover, it would be interesting to compare the performance and the preference of the novices with the ones of trained operators. The previous knowledge might affect it positively by making it faster in reaching the plateau of the learning curve, as shown in previous studies [28], or negatively affected it due to the mismatch in the movements to manipulate the instruments. The preference is expected to match the novice's preference due to the similarities with the two DOF laparoscopic instruments currently used in the field. Another aspect that would

be interesting to further investigate is the possible applications of our instruments by using the available lumens to insert flexible instruments to grasp tissues or perform biopsy procedures.

By comparing two cable configurations in 3D printed steerable instruments, this study explores new possibilities for additive manufacturing technology in medical instruments where complex geometries for the single parts simplify the overall design while maintaining, if not enhancing, the instrument's functionalities.

## 7. Conclusion

The goal of this study was to compare parallel and multi cable configurations in multi-steerable laparoscopic instruments in terms of task performance time and workload. Our experiment showed that there was no significant difference in the task performance time for the two configurations. In the used NASA TLX scale, however, the participants expressed a lower workload for the parallel configuration as compared to the multi configuration. Overall, 10 out of 12 participants preferred the parallel configuration. The preference was mainly determined by the increased possibility of individually orienting the most distal segment.

## Supporting information

**S1 Fig. Flow chart of the experiment for each participant.** Each trial consists of 12 runs and the order of the instruments used for the two groups. Parallel configuration (PC), multi configuration (MC).
(TIF)

**S1 Video. Video of the execution of one run with the parallel configuration in strument and one run with the multi configutation instrument.**
(MP4)

## Author Contributions

**Conceptualization:** Costanza Culmone, Remi van Starkenburg, Gerwin Smit, Paul Breedveld.

**Data curation:** Costanza Culmone.

**Formal analysis:** Costanza Culmone.

**Funding acquisition:** Paul Breedveld.

**Investigation:** Costanza Culmone.

**Methodology:** Costanza Culmone, Remi van Starkenburg, Gerwin Smit, Paul Breedveld.

**Supervision:** Gerwin Smit, Paul Breedveld.

**Validation:** Costanza Culmone.

**Writing – original draft:** Costanza Culmone.

**Writing – review & editing:** Remi van Starkenburg, Gerwin Smit, Paul Breedveld.

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
