## [Decision Letter · Decision Letter 0]

6 Jul 2022

PONE-D-22-13417Comparison of two cable configurations in 3D printed steerable instruments for Minimally Invasive SurgeryPLOS ONE

Dear Dr. Culmone,

Thank you for submitting your manuscript to PLOS ONE. After careful consideration, we feel that it has merit but does not fully meet PLOS ONE’s publication criteria as it currently stands. Therefore, we invite you to submit a revised version of the manuscript that addresses the points raised during the review process.

Major concerns provided by the reviewers are related to the motivation and significance of the study as well as the statistical analyses. Please, address all the comments made by the reviewers.

We look forward to receiving your revised manuscript.

Kind regards,

Antonio Riveiro Rodríguez, PhD

Academic Editor

PLOS ONE

Journal Requirements:

Reviewers' comments:

Reviewer's Responses to Questions

**Comments to the Author**

1. Is the manuscript technically sound, and do the data support the conclusions?

Reviewer #1: Yes

Reviewer #2: Yes

Reviewer #3: Yes

2. Has the statistical analysis been performed appropriately and rigorously? 

Reviewer #1: Yes

Reviewer #2: Yes

Reviewer #3: No

3. Have the authors made all data underlying the findings in their manuscript fully available?

Reviewer #1: Yes

Reviewer #2: Yes

Reviewer #3: Yes

4. Is the manuscript presented in an intelligible fashion and written in standard English?

Reviewer #1: Yes

Reviewer #2: Yes

Reviewer #3: Yes

5. Review Comments to the Author

Reviewer #1: This work is interesting and clearly shows the results of their research. However, I have some comments and doubts about this study:

1.- Why was a direct view used instead of a 2D view on a monitor? Although this increases the complexity of the trials, it is necessary to recreate all the conditions of the minimally invasive procedures to obtain an idea of what is happening.

2.- Do the cables that attach to the links of the instrument suffer from any fatigue? How often did you change all the cables or the instrument? Every 3 or 4 attempts? Understanding that many articulating instruments suffer from fatigue or lack of tension in the cable arrangement after 10 attempts, it would be interesting to comment on this point of the investigation.

3.- Although no differences were shown between the two configurations, parallel and multi, would crossing the cables allow to cancel the fulcrum effect at the tip of the instrument controlled by the handle?

Reviewer #2: The authors compared two cable configurations for laparoscopy instruments fabricated via 3D printing. The topic is interesting, the paper is well-written and free from significant flaws. However, I suggest to accept it only after minor revisions. There are three points that can be improved:

1) In general, the authors should clearly indicate all the dimensions of the devices presented in the paper.

2) Section 3.2: it would be interesting to add more details on the 3D printing process. For example, the authors should provide the 3D models and the exact dimensions of the 3D printed parts that constitute the devices. How were these parts printed? Did the authors use printing supports? Did they postprocessed the 3D printed parts?

3) Section 4.2: can the authors comment on the choice of recruiting participants with no experience in laparoscopy? Is it expectable a difference between untrained and trained operators?

Reviewer #3: What is the size of the instruments? Scales (diameter, length, etc.) are missing in the figures.

For the multi configuration, how were the radii determined to avoid overlapping the cables?

Cable configuration in cable-driven flexible devices is of an issue in manipulation and control. However, there are many other scientific ways (some studied by same group of authors) to tackle the problem rather than user’s intuitive performance.

Motivations for the study is not well laid out. Why is this study needed, and where is the significance? Does the study intent to present superiority of one design over another?

What did this study aim to confirm or conclude?

Statistical analyses to decide the number of participants and number of tests are not provided. Without this, no conclusions can be drawn.

The learning curve could be observed with unexperienced participants. Clarification needed on how workload and learning curve were quantified.

It is intended that these tools will eventually be used in the hands of experienced surgeons, so what preference is shown by surgeons?

What criteria were taken into consideration designing the tests, repetitions, etc.

How was the data analyzed?

There was a preference for the parallel config because the distal joint could be oriented more reliably. There is no reason given for why this was important in the context of the experiment though...In the video it appears that the tip orientation needed to be very precise in order to activate the target- it was likely harder to align the tip with such precision by using the MC, but it appeared that the initial positioning to reach the target (before the fine-tuning motions) using the MC was faster. How does that align with real laparoscopic practices? Would the effect of a small lack of precision in the tip angle be negligible or drastic for surgeons?

Clarification needed on “amplification component”

Figure 8 is unnecessary

Figure 9: please choose more contrasting colors for your plots. It’s difficult to see the difference between PC and MC plots

Figure 10: Trial is incorrectly spelt

Fig. 2a: Orientation angle is mentioned, but not shown.

Fig. 2b: All used parameters in that figure should be explained (alpha is missed)

In center parts of Fig. 2, the vertical axis is not named.

Lines 193 to 208 discuss differences in control and segments, but the purpose/result is not clear.

Clinical application: The materials and manufacturing needs improvement and the device presented is only a prototype – this research is maybe at too early for user trials.

6. PLOS authors have the option to publish the peer review history of their article (what does this mean?). If published, this will include your full peer review and any attached files.

Reviewer #1: No

Reviewer #2: No

Reviewer #3: No

---

## [Author Response · Author response to Decision Letter 0]

2 Sep 2022

Dear Editor and Reviewers,

We would like to thank you for the efforts in reading our manuscript and providing useful feedback. We have carefully gone through the reviewers’ comments and created an improved version of the paper. We are glad to resubmit the new manuscript for evaluation, together with this accompanying letter answering in detail to each reviewer comment. All the modifications introduced in the revision process are highlighted in red in the text.

Kind regards,

The Authors

Reviewer 1

This work is interesting and clearly shows the results of their research. However, I have some comments and doubts about this study:

Comment 1

Why was a direct view used instead of a 2D view on a monitor? Although this increases the complexity of the trials, it is necessary to recreate all the conditions of the minimally invasive procedures to obtain an idea of what is happening.

Authors

We would like to thank the reviewer for this comment. We choose of using a transparent phantom to address the difficulties that users with no knowledge of laparoscopic procedures might encounter in properly visualizing the operating space through a 2D view on a monitor. Differently, by using laparoscopic view settings, the collected data would have reflected the learning curve of the participants in visualizing the space in 3D from a 2D image, more than the capacity to properly operate the device. 2D visualization would instead be recommended when testing the device with experts. For clarity, we added this consideration in Section 4.3 Experiment setup, lines 286-288.

Comment 2 

Do the cables that attach to the links of the instrument suffer from any fatigue? How often did you change all the cables or the instrument? Every 3 or 4 attempts? 

Understanding that many articulating instruments suffer from fatigue or lack of tension in the cable arrangement after 10 attempts, it would be interesting to comment on this point of the investigation.

Authors

We thank the reviewer for this comment. The cables were preliminarily tested before the experiments to check if they were suffering fatigue. The test did not show any particular fatigue when straightened. Once mounted on the device, cables were never changed during each individual test. Therefore, they were able to withstand a total of 48 attempts. We noticed that, due to the space between the helical elements of the handle, the cables showed possible local bends during the use of the instrument. We believe that the bending was due to the excessive force applied by the participant during the first attempts, while trying to familiarize themselves with the devices. When the instrument was used with the appropriate force, no bending or fatigue was noticed. All participants used the same device from the beginning to the end of the test, except for one participant for who damaged the device end-effector by applying excessive force. We have now added these details for clarity in the Discussion, lines 490-497.

Comment 3

Although no differences were shown between the two configurations, parallel and multi, would crossing the cables allow to cancel the fulcrum effect at the tip of the instrument controlled by the handle?

Authors

We would like to thank the reviewer for the comment. All cables for both the multi and the parallel configuration run from the tip's distal end to the proximal end of the handle. Rotating the cables 180 degrees in the shaft would let the tip move in the opposite direction, mirroring the current tip movement. However, it will not eliminate the fulcrum effect caused by the rigid shaft, pivoting in the abdominal wall.

Reviewer 2 

The authors compared two cable configurations for laparoscopy instruments fabricated via 3D printing. The topic is interesting, the paper is well-written and free from significant flaws. However, I suggest to accept it only after minor revisions. There are three points that can be improved:

Comment 1

In general, the authors should clearly indicate all the dimensions of the devices presented in the paper.

Authors

We would like to thank the reviewer for this suggestion. We have now reported the dimensions of the devices in details in Figure 3. 

Comment 2

Section 3.2: it would be interesting to add more details on the 3D printing process. For example, the authors should provide the 3D models and the exact dimensions of the 3D printed parts that constitute the devices. How were these parts printed? Did the authors use printing supports? Did they postprocessed the 3D printed parts?

Authors

We would like to thank the reviewer for this comment. The 3D model of the two devices is used in Figure 3 to present the parallel and multi configurations. We improved the caption to clarify and refer to them as the 3D model. Moreover, we implemented Section 3.2 Fabrication with an improved description of the printing process: “The shaft was printed without any support except for the raft. The handle instead, was printed with the raft, and the internal support was made of small pillars autogenerated by the printer software. After the printing, the excess resin was removed by placing the parts in an isopropyl alcohol bath for 30 minutes. The support of the handle and the raft were manually removed. The dimensions of the printed devices slightly differ from the CAD model with a tolerance of � 0.08mm for the shaft and � 0.12 mm for the handle”. The integration can be found on lines 227-232.

Comment 3

Section 4.2: can the authors comment on the choice of recruiting participants with no experience in laparoscopy? Is it expectable a difference between untrained and trained operators?

Authors

We would like to thank the reviewer for this comment. The experiments were carried out in March 2021 when restrictions due to the pandemic were still tight. For this reason, access to hospitals, and therefore recruiting subjects with experience in laparoscopy, was not possible. We recognize how comparing the performance between novices and experienced operators, would represents a valuable follow up study. Two different outcomes could be expected: compared to novice users, previous experience with different laparoscopic instruments (most of them with 2 degrees of freedom at the end effector) could constitute an initial obstacle, making the learning curve slower in the first trials, or could help them instead to reach faster a plateau in the learning curve, as shown in previous studies on articulating laparoscopic instruments “The trade-off between flexibility and maneuverability: task performance with articulating laparoscopic instruments”, Martinec et al., (2009). However, independently from previous experience, we would still expect a strong preference for the parallel configuration due to the similarities in orientation of the conventional steerable laparoscopic instruments. We elaborated on this point in Section 6.2, lines 499-506. 

Reviewer 3

Comment 1

What is the size of the instruments? Scales (diameter, length, etc.) are missing in the figures.

Authors

We would like to thank the reviewer for this comment. We have now indicated in details the dimension of the device in Figure 3.

Comment 2

For the multi configuration, how were the radii determined to avoid overlapping the cables?

Authors

We thank the reviewer for this comment. The radii to avoid the overlap of the cables in the multi configuration were determined considering the printability of the instrument and the cable size. Once decided the amplification factor by fixing the diameter of the handle, the cables were placed as close as possible to each other but at the same time kept independent with dedicated grooves. Moreover, the order of the cables (clockwise, counterclockwise, and parallel) was decided considering the easiest way to mount the cables into the instrument. We improved this part in the Section 2.3, lines 156-157, and Section 3.2, lines 233-238. 

Comment 3

Cable configuration in cable-driven flexible devices is of an issue in manipulation and control. However, there are many other scientific ways (some studied by same group of authors) to tackle the problem rather than user’s intuitive performance. Motivations for the study is not well laid out. Why is this study needed, and where is the significance? 

Authors

We would like to thank the reviewer for this comment. Fully mechanical cable-driven instruments have the advantage to have low maintenance costs, they make no noise, they have high sensitivity, and high speed and they directly react to the surgeon’s movement giving the haptic feedback that is missing in many robotic solutions, but more importantly, they allow simplification of the design without compromising the instrument functionality. For this reason, cable-driven mechanisms still represent a valid alternative for laparoscopic instruments. We improved the Introduction in order to highlight this point on lines 66-69.

Comment 4

Does the study intent to present superiority of one design over another? What did this study aim to confirm or conclude?

Authors

We thank the reviewer for this comment. The study does not present the superiority of a design over another, but rather highlight and compare important weak and strong point of the new designs, that can help to improve the design of future multi-steerable laparoscopic instruments. For example, the performed tests showed a net preference for the parallel configuration over the multi configuration, proving that the independent orientation of the end-effector is a fundamental factor for subjects with no experience in steerable laparoscopic instruments. However, the multi configuration appeared to be faster in positioning the instrument to reach the target We improved the goal of the work in order to clarify the scope of the study, on lines 82-83 of the Introduction.

Comment 5

Statistical analyses to decide the number of participants and number of tests are not provided. Without this, no conclusions can be drawn.

Authors

We would like to thank the reviewer for this comment. We agree with the reviewer on the importance of statistical analysis to decide the number of participants. However, due to the difficulties in defining the target population and the number of instruments used in the field being the instrument new in such application, we decided to base the number of participants for this study on similar studies i.e., “SATA-LRS: A modular and novel steerable hand-held laparoscopic instrument platform for low-resource settings” Lenssen et al., (2022), “The trade-off between flexibility and maneuverability: task performance with articulating laparoscopic instruments”, Martinec et al., (2009), “Comparison of Laparoscopic Steerable InstrumentsPerformed by Expert Surgeons and Novices”, Lacitignola et al., (2020), “Precision in stitches: Radius Surgical System”, Waseda, (2007), where it varies between 5 and 24. In these studies either novices, experts, or both of the categories are recruited as participants of the study. Moreover, among the 12 participants of the studies, we tried to have differences such as age or gender and different habits that could affect the performances such as playing instruments or video games. For what concerns the number of tests, a similar consideration can be drawn. We decided to organize the test in four trials per participant so that each participant could test each instrument two times and the two Groups A and B could be comparable. Moreover, we fixed the number of repetitions to 12 so that the participants had the time to first familiarized themselves with the instrument and then improve his/ her performance. A similar number of attempts have also been used by similar studies such as SATA-LRS: A modular and novel steerable hand-held laparoscopic instrument platform for low-resource settings” Lenssen et al., 2022", where the study has been conducted with novices, whereas a lower number of attempts has been used in studies that involved also expert surgeons, such as “Comparison of precision and speed in laparoscopic and robot-assisted surgical task performance” Zihni et al. (2017). We included this consideration in Section 4.2 266, 268-271, and Section 4.3 lines 315-318.

Comment 6

The learning curve could be observed with unexperienced participants. Clarification needed on how workload and learning curve were quantified.

Authors

The learning curve was quantified for two different aspects. First, it was calculated as the difference between the average time used to perform the first run and the one to perform the last run for each instrument in the two different trials. Then the learning curve was calculated as the difference between the two groups over the four performed trials. The different drop in time is reported in percentage in the study. The workload was quantified using the NASA-TLX test which is generally used to quantify the workload of a task. The test is a self-evaluation that the participants had to fill out at the end of each trial. In the form, the participant had to mark on a scale from very low to very high how demanding the test was. The results have been evaluated by considering very low as 0 % and very high as 100 % workload. The results of the test for each trial have been plotted in Figure 12 (new Figure 11). The code used to analyze all data is available in the link of “data availability”. We improved the test to make these points clear in Section 5 lines 365-366, 388-390, and 399.

Comment 7 

It is intended that these tools will eventually be used in the hands of experienced surgeons, so what preference is shown by surgeons?

Authors

We would like to thank the reviewer for this comment. Due to the fact that the proposed instruments have not been tested by experienced operators, we can only make some considerations for what concern the possible scenarios. Either the previous experience with different laparoscopic instruments, most of them with none or 2 degrees of freedom at the end effector, would constitute an initial obstacle and would make the learning curve slower than for the novices in the first trials, or their knowledge would help them and the learning curve would reach a plateau faster than for novices, as it has been shown in previous studies when a comparison with conventional and articulating laparoscopic instruments was conducted with experts and novices (The trade-off between flexibility and maneuverability: task performance with articulating laparoscopic instruments, Martinec, 2009). The authors would expect a similar outcome for what concerns the type of instruments with a strong preference for the parallel configuration due to the similarities in the orientation of the conventional steerable laparoscopic instruments. We elaborated on this point in Section 6.2, lines 501-506. 

Comment 8

What criteria were taken into consideration designing the tests, repetitions, etc.

How was the data analyzed?

Authors

We would like to thank the reviewer for this comment. The test was designed considering that the steerable laparoscopic instruments compared to the conventional laparoscopic instruments allow more precise positioning of the instruments and orientation of the end-effector. Therefore, in the designed platform each tube has a different orientation and position that can be reached only with a specific shape of the end-effector. The repetitions of the trials were decided based on the fact that each instrument had to be used two times to have a comparison among the two groups and, in case, notice differences or influences of one instrument over the other. For each run, all the targets need to be randomly reached so that all runs could be comparable. Each trial has 12 runs so that the participants would during the first runs familiarize themselves with the instrument and then try to improve his/her performance. Moreover, a comparable number of attempts has been used in similar studies such as “SATA-LRS: A modular and novel steerable hand-held laparoscopic instrument platform for low-resource settings” Lenssen et al., 2022", where the study has been conducted with novices, whereas a lower number of attempts has been used in studies that involved also expert surgeons, such as “Comparison of precision and speed in laparoscopic and robot-assisted surgical task performance” Zihni et al. (2017). Data have been analyzed using the Matlab script available in the data availability of this article. We now added this consideration to clarify in Section 4.3 lines 315-318, and 336.

Comment 9

There was a preference for the parallel config because the distal joint could be oriented more reliably. There is no reason given for why this was important in the context of the experiment though...

Authors

We would like to thank the reviewer for this comment. During surgery precision and accuracy are extremely important to guarantee the success of the procedure. Having a fast and reliable orientation of the surgical instrument allow reaching the target while preserving critical surrounding areas. Therefore, in the experiment, we aimed at recreating in simulating environment the movement that the surgeon might perform when navigating during a laparoscopic procedure with particular importance to the precision in the orientation of the instrument. We improved Section 4.3 in lines 278-281.

Comment 10

In the video it appears that the tip orientation needed to be very precise in order to activate the target- it was likely harder to align the tip with such precision by using the MC, but it appeared that the initial positioning to reach the target (before the fine-tuning motions) using the MC was faster. How does that align with real laparoscopic practices? Would the effect of a small lack of precision in the tip angle be negligible or drastic for surgeons?

Authors

We would like to thank the reviewer for this comment. When using conventional laparoscopic instruments, the surgeons lose dexterity, and, therefore, precision becomes also a challenge. The use of steerable laparoscopic instruments allows the surgeon to be more precise and accurate, as shown in previous studies by Weseda et al. "Precision in stitches: Radius Surgical System". In our case, the lack in the precision of the MC would probably not be an issue for standard procedures, as it can be compared to commercialized steerable laparoscopic instruments, however, the high precision of the PC might allow procedures that are nowadays not possible due to such a lack of precision. We included this consideration in the Discussion, Section 6.1, lines 463-464 and 470-473.

Comment 11

Clarification needed on “amplification component”

Authors

We would like to thank the reviewer for this comment. We specified in the text that the “amplification is the distal rigid part of the handle (Section 3.1 line 191), and we improved Figure 3 so that also the amplification component is named in it.

Comment 12

Figure 8 is unnecessary

Authors

We would like to thank the reviewer for this comment. We moved Figure 8 to the supplementary materials. 

Comment 13

Figure 9: please choose more contrasting colors for your plots. It’s difficult to see the difference between PC and MC plots

Authors

We would like to thank the reviewer for this comment. We changed the color that represents the parallel configuration from green to yellow in order to improve the contrast. All figures and test have been changed accordingly.

Comment 14

Figure 10: Trial is incorrectly spelt

Authors

We would like to thank the reviewer for this comment. We corrected the spelling of Trial in the previous Figure 11, now Figure 10.

Comment 15

Fig. 2a: Orientation angle is mentioned, but not shown.

Authors

We thank the reviewer for this comment. We included the orientation angle in Figure 2. We named it beta in the text and in the figure. 

Comment 16

Fig. 2b: All used parameters in that figure should be explained (alpha is missed)

Authors

We thank the reviewer for this suggestion. We added the explanation of the alpha parameter in the text, line 114. 

Comment 17

In center parts of Fig. 2, the vertical axis is not named.

Authors

We thank the reviewer for this comment. We added the name of the vertical axis in the central parts of Fig 2. 

Comment 18

Lines 193 to 208 discuss differences in control and segments, but the purpose/result is not clear.

Authors

We would like to thank the reviewer for this consideration. We improved Section 3.1 (lines 215-218) by explaining that, due to the equal length of the steerable shaft, the two prototypes are able to cover the same workspace. Moreover, the identical appearance of the two prototypes allows for comparing the cable configuration performance without influencing the participant in the test.

Comment 19

Clinical application: The materials and manufacturing needs improvement and the device presented is only a prototype – this research is maybe at too early for user trials.

Authors

We would like to thank the reviewer for this comment. We agree with the reviewer about the early stage of such research to directly be used in a trial. In this work, we presented a first evaluation of the maneuverability of our 3D-printed steerable instruments. The presented test is fundamental, especially for inexperienced subjects to have an immediate impression of the possible use of such instruments. In case of complete failure of our studies with subjects that would have given up in the middle of the test or an increase of time during different trials would have suggested a possible need for modification of our approach in investigating 3D printed steerable instruments. On the contrary, the feedbacks were all positive and the subjects enjoyed participating in the training. We implement this consideration in the Discussion, Section 6.1, lines 473-476.

---

## [Decision Letter · Decision Letter 1]

19 Sep 2022

Comparison of two cable configurations in 3D printed steerable instruments for Minimally Invasive Surgery

PONE-D-22-13417R1

Dear Dr. Culmone,

We’re pleased to inform you that your manuscript has been judged scientifically suitable for publication and will be formally accepted for publication once it meets all outstanding technical requirements.

Kind regards,

Antonio Riveiro Rodríguez, PhD

Academic Editor

PLOS ONE

Reviewers' comments:

Reviewer's Responses to Questions

**Comments to the Author**

1. If the authors have adequately addressed your comments raised in a previous round of review and you feel that this manuscript is now acceptable for publication, you may indicate that here to bypass the “Comments to the Author” section, enter your conflict of interest statement in the “Confidential to Editor” section, and submit your "Accept" recommendation.

Reviewer #1: All comments have been addressed

Reviewer #2: All comments have been addressed

2. Is the manuscript technically sound, and do the data support the conclusions?

Reviewer #1: Yes

Reviewer #2: Yes

3. Has the statistical analysis been performed appropriately and rigorously? 

Reviewer #1: Yes

Reviewer #2: Yes

4. Have the authors made all data underlying the findings in their manuscript fully available?

Reviewer #1: Yes

Reviewer #2: Yes

5. Is the manuscript presented in an intelligible fashion and written in standard English?

Reviewer #1: Yes

Reviewer #2: Yes

6. Review Comments to the Author

Reviewer #1: I consider that the authors have made the necessary changes to publish the document; therefore, my decision is to accept the article.

Reviewer #2: The authors satisfactorily replied to all my observations. In my opinion, the paper can now be published in its current form.

7. PLOS authors have the option to publish the peer review history of their article (what does this mean?). If published, this will include your full peer review and any attached files.

Reviewer #1: No

Reviewer #2: No

---

## [Editor Report · Acceptance letter]

26 Sep 2022

PONE-D-22-13417R1 

Comparison of two cable configurations in 3D printed steerable instruments for Minimally Invasive Surgery 

Dear Dr. Culmone:

I'm pleased to inform you that your manuscript has been deemed suitable for publication in PLOS ONE. Congratulations! Your manuscript is now with our production department. 

Kind regards, 

on behalf of

Dr. Antonio Riveiro Rodríguez 

Academic Editor

PLOS ONE